# Alkylation of Aromatic Compounds with Pentabromobenzyl Bromide and Tetrabromoxylene Dibromide as a New Route to High Molecular Weight Brominated Flame Retardants

**DOI:** 10.3390/polym12020352

**Published:** 2020-02-06

**Authors:** Mark Gelmont, Michael Yuzefovitch, David Yoffe, Eyal Eden, Sergei Levchik

**Affiliations:** 1IMI Institute for R&D, Haifa Road, Industrial Zone Kiryat Ata, Haifa Bay 28226, Israel; mark.gelmont@icl-group.com (M.G.); michael.yuzefovitch@icl-group.com (M.Y.); david.yoffe@icl-group.com (D.Y.); 2ICL-IP, P.O. Box 180, Beer Sheva 84101, Israel; eyal.eden@icl-group.com; 3ICL-IP America, 769 Old Saw Mill River Rd, Tarrytown, NY 10591, USA

**Keywords:** flame retardant, pentabromobenzyl bromide, tetrabromoxylylene dibromide, polypropylene, polyamide 6.6, HIPS, ABS

## Abstract

In the view of many national and international human health and environmental regulations, polymeric flame retardants are sustainable products. In this work, a series of high molecular weight and polymeric brominated flame retardants are synthesized by the alkylation of aromatic molecules or the alkylation of aromatic polymers with pentabromobenzyl bromide (PBBB) or tetrabromoxylylene dibromide (TBXDB). The flame retardants prepared via the alkylation of toluene or diphenylethane with PBBB were found to be not truly polymeric but had high *M*_w_ > 1400. However, the alkylation of the same aromatic molecules by a mixture of PBBB and TBXDP resulted in polymeric flame retardants with *M*_w_ > 130,000. Two other polymeric flame retardants were prepared by the alkylation of aromatic polymers (polyphenylene ether or polystyrene) with PBBB. It was found that the new flame retardants had a high bromine content of more than 68%. They showed high thermal stability with the onset of thermal decomposition above 360 °C and a maximum rate of weight loss at about 375–410 °C. The newly synthesized flame retardants were tested in different thermoplastics. Flame retardant efficiency and physical properties were comparable or better than the reference commercial flame retardants.

## 1. Introduction

Because a bromine atom is heavy, brominated flame retardants (FRs) have larger molecular weight (typically > 700 g/mole) compared to other types of organic flame retardants. From a regulatory point of view molecules consisting of at least three repeating units and having molecular weight > 1500 are recognized as polymers in most geographical areas. Polymeric flame retardants have a few obvious technical, environmental, and toxicological benefits. However, it is important to note that there are some brominated flame retardants with molecular weight > 1500 g/mole that formally cannot be called polymers because they are not built with repeating units.

Due to the nature of biological membranes very large molecules cannot pass through them. This phenomenon is well known in pharmacology as the “Liminski Rule of Five” which is used to predict biological activity of chemicals based on molecular weight and lipophilicity [1]. The rule suggests that chemicals with molecular weight > 500 g/mole exhibit poor permeation of cell membranes. Thus, the membranes of the skin, lungs, and gastrointestinal tract typically do not allow large molecules to pass through. If large molecular weight brominated FR is ingested, it is usually excreted without being substantially absorbed into the body [2]. Because of this polymeric and large non-polymeric molecules avoid the problem of several types of lower molecular weight brominated flame retardants that are the subject of restrictions in use [3].

Polymeric FRs typically have low solubility in water, which means that once incorporated into the end-product plastic matrix, they become integrated with the plastic and leaching is unlikely to occur. Because of negligible vapor pressure they do not migrate to the surface of the plastic during aging, thus significantly diminishing or eliminating any potential blooming in the finished product. Extensive testing has also shown that formation of polybrominated p-dibenzodioxins and dibenzofurans can be controlled by proper incineration of thermoplastics containing polymeric brominated FRs [4,5]. It has also been found that thermoplastics containing polymeric brominated FRs show high thermal stability and allow good physical properties of re-processed polymers in mechanical recycling operations to be maintained [6,7,8,9]. This is a very important consideration from an environmental viewpoint.

For many years the flame retardant industry has produced a series of polymeric brominated flame retardants. The most common are brominated polystyrene, which is predominantly used in polyamides; brominated polycarbonate and poly(pentabromobenzyl acrylate), which are mostly used in thermoplastic polyesters; and brominated epoxy polymers, which have a wide range of applications from styrenic thermoplastics to engineering resins [10]. A most recent addition to the family is brominated styrene–butadiene–styrene block copolymer [11] (only the butadiene part is brominated) which finds application in polystyrene foam insulation [12]. Because compatibility (miscibility) of polymers depends very much on polymer structure and molecular weight, the industry is actively searching for new brominated flame retardants [13,14,15,16,17] which are sustainable and can replace traditional non-polymeric counterparts.

In this paper, we synthesize new compounds containing pentabromobenzyl moieties. These compounds are prepared by electrophilic C-alkylation of different aromatic compounds with pentabromobenzyl bromide (PBBB) in the presence of Friedel-Crafts catalysts. The pentabromobenzyl-group-containing compounds have a high molecular weight (>1000) and a high bromine content (not less than 70%). They are insoluble in water, stable against hydrolysis, and show very high thermal stability. Using a similar synthetic route, we also perform C-alkylation using difunctional tetrabromoxylylene dibromide (TBXDB), which results in polymeric products. The alkylation with both PBBB and TBXDB leads to high bromine content polymeric products with pentabromobenzyl pendant groups. Finally, we alkylate polyphenylene ether (PPE) and polystyrene (PS) with PBBB. Because PPE is an oligomeric and PS is a polymeric precursor, the obtained brominated flame retardants are polymers as well.

## 2. Materials and Methods

### 2.1. Materials

Technical grade bromine from ICL-IP (Beer Sheva, Israel) was used for the bromination. Toluene and xylene were purchased from Bio-Lab Chemicals (Jerusalem, Israel). Pentabromobenzyl bromide (PBBB) was prepared according to the known method described in the patent literature [18,19]. In short, toluene was first reacted with elemental bromine in the presence of an AlCl_3_ catalyst and using dibromomethane as a solvent, which resulted in the bromination of the aromatic ring. In the second step, the methyl group was brominated using elemental bromine and AIBN as a free radical initiator. TBXDB was prepared by a similar synthesis route. Diphenylethane used in further alkylation reactions was provided by Bailly (Taizhou, China). Polyphenylene ether (Santovac 7) of molecular weight (*M*_w_) ~450 corresponding to about five phenyl units was purchased from Santolubes LLC. (Spartanburg, SC, USA) Polystyrene *M*_w_ = 170,000 was supplied by Arkema (Colombes, France).

### 2.2. Chemical Analysis

The bromine content of the new flame retardants was measured by the Parr Bomb method. The sample (~0.08–0.12 g) was placed in a peroxide bomb vessel, after which 0.5 g of sucrose was added; finally, a full dipper of sodium peroxide was added. The bottom of the bomb was heated up to about 200 °C with a Bunsen burner so that the sample was fully oxidized by sodium peroxide. Then, the bomb was cooled down and the residual pressure was released. Bromine mostly in the form of sodium bromide was retained in the bomb. The content of the bomb was then combined with warm water and hydrazine sulfate was added to destroy residual sodium peroxide. Nitric acid was added in portions until the solution became slightly acidic. The solution was cooled down and titrated with AgNO_3_ (0.1 N) to determine bromine content.

### 2.3. Gel Permeation Chromatography (GPC)

The molecular weight of the new polymeric flame retardants was measured using a Viscotek HT-GPC 350A module (Malvern Instruments, Malvern, UK) equipped with an RI, viscometer, and light scattering detectors. Two Tosoh TSK-GEL GMHhr-H(S) HT 7.8 mm × 30.0 cm, 13 µm GPC columns were used. The samples were dissolved in 1,2,4-trichlorobenzene and stabilized with 2,6-di-tert-butyl-4-methyl-phenol (BHT). The flow rate was set to 1.0 mL/min and the temperature of both the detector and column was set to 120 °C. Calibration was done using a Malvern TDS2000 (HT-TDS-PS) polystyrene standards kit.

### 2.4. Infrared Spectroscopy (FTIR)

Infrared spectra of new flame retardants were collected on a Bruker Tensor II FTIR instrument (Billerica, MA, USA) using the ATR technique.

### 2.5. Thermal Analysis

TGA analysis was performed using TGA Q500, TA Instruments (New Castle, DE, USA). Samples of ~10 mg were heated in an aluminum oxide crucible from 35 to about 700 °C with a heating rate of 10 °C/min in a nitrogen atmosphere.

### 2.6. Ingredients, Compounding, and Molding

Polymers and other components of the flame retardant compositions are listed in Table 1. A Berstorff ZE25 co-rotating twin screw extruder (KraussMaffei Berstoff, Hanover, Germany) with L/D = 32 was used to compound the newly synthesized flame retardants in different polymers. Commercial flame retardant additives listed in Table 1 were used as benchmarks in different polymers. Polypropylene, high-impact polystyrene (HIPS) and acrylonitrile–butadiene–styrene copolymer (ABS) were used as received whereas polyamide (PA) 6.6 was dried in a vacuum oven at 80 °C overnight before extrusion. The polymer, the flame retardants, and all other ingredients were thoroughly mixed in a plastic bag and fed into the main port using a volumetric feeder. In the case of PA 6.6 glass fibers were fed into heating zone 3 using the side feeder. Polypropylene was extruded at 160–230 °C, HIPS at 180–220 °C, ABS at 180–240 °C, and PA 6.6 at 250–280 °C. The strands produced in the extruder were cooled under water and pelletized using an Accrapak System 750/3 pelletizer (Accrapak Sytems, Warrington, UK). Obtained pellets were dried in a circulating air oven at 80 °C for at least 3 h before they were injection molded into the test specimens using an Allrounder 500–150 molding machine (Arburg, Lossburg, Germany). The test specimen 1.6 mm standard UL-94 bars and standard “dog-bone” bars for physical testing were conditioned at 23 °C for one week before testing.

### 2.7. Flammability Test

A flammability test was carried out according to the Underwriters-Laboratories standard UL-94, applying a vertical Bunsen burner to the standard specimens of 1.6 mm thickness for polypropylene, ABS, and HIPS and 0.8 mm for glass-filled PA 6.6.

### 2.8. Mechanical Properties

Izod notched impact strength was measured using a CEAST 9050 Pendulum Impact System (Instron, Norwood, MA, USA) following the ASTM D-256 protocol. Tensile properties (tensile strength, tensile modulus, and elongation at break) were measured using a Z010 Material Tester (Zwick Roell, Ulm, Germany) according to ASTM D-638 at a 5 mm/min speed of clamp using type 2 dumbbells.

### 2.9. Heat Distortion Temperature (HDT)

HDT was measured according to ASTM D648 at a heating rate of 2 °C/min and a 1820 KPa cell load using a HDT/Vicat-Plus (Lloyd Instruments, West Sussex, UK).

### 2.10. Melt Flow Index (MFI)

MFI was determined according to ASTM D1238 using a Meltflixer 2000 melt flow instrument (Thermo Hake, Karlsruhe, Germany) with a cell load of 2.16 kg. A temperature of 230 °C was used for polypropylene, HIPS, and ABS and 250 °C for PA 6.6.

### 2.11. Synthesis

#### 2.11.1. Alkylation of Toluene with PBBB (T-PBBB)

Dibromomethane (200 mL), PBBB (62.2 g, 0.11 mol), and toluene (3.7 g, 0.04 mol) were placed into a 500 mL flask fitted with a mechanical stirrer, thermometer, condenser and N_2_ inlet. The mixture was heated to 70 °C until the PBBB was dissolved. AlCl_3_ (0.7 g, 0.005 mol) was added and the vigorous formation of HBr was started. The mixture was heated at 80 °C for 6 h until the PBBB disappeared (controlled by GC). The reaction mixture was washed three times with water (3 × 120 mL) and sodium bisulfite (1.5 mL, ~28% aqueous solution), with 20 min taken for each washing. After that, the solid was filtered out and re-slurried with dichloromethane (2 × 200 mL) at 40 °C for one hour (each re-slurry). The reaction mixture was cooled to 20 °C and the solid was filtered off and dried in an oven at 150 °C under reduced pressure for 24 h to give 42.7 g, corresponding to an ~75% yield, based on PBBB. According to the elemental analysis, the content of bromine was about 76%, corresponding to ~2.7 PBBB units per one molecule of toluene. The estimated molecular weight was about 1400. The FTIR spectrum of T-PBBB is shown in Figure 1. The major absorption bands attributed to the pentabromobenzene ring are 1316 cm^−1^ (semicircle stretching), 1060 cm^−1^ (C–Br stretching), and 552 cm^−1^ (ring bending) [20]. The bands at 2980 cm^−1^ (CH_3_ stretching), 1431 cm^−1^ (semicircle stretching) and 947 cm^−1^ (aromatic C–H bending) can be assigned to substituted toluene. Solubility in organic solvents of P-PBBB was very poor, which did not allow for the performing of NMR analysis. The chemical structure of P-PBBB can be represented as follows. 
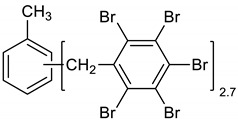
(1)

#### 2.11.2. Alkylation of Diphenylethane with PBBB (DPE-PBBB)

Dichloroethane (1600 mL), PBBB (805.6 g, 1.42 mol) and diphenylethane (57.70 g, 0.317 mol) were placed into a 2000 mL glass reactor fitted with a mechanical stirrer, thermometer, condenser, and N_2_ inlet. The mixture was heated to 70 °C and AlCl_3_ (4.5 g, 0.17 mol) was added portion wise over ~3 h. Then, the mixture was heated for an additional hour at 65–75 °C until the PBBB disappeared (by GC). The reaction mixture was washed three times with water (3 × 1000 mL) at ~60 °C and sodium bisulfite (20 mL, ~28% aqueous solution), with 20 min taken for each washing. After this, the solid was filtered off at 40–50 °C, washed with 200 mL dichloroethane, and dried in an oven at 150 °C under reduced pressure for 20 h to give 738 g, corresponding to an ~98% yield. The content of bromine was ~75%. The product contained on average five PBBB units and had a *M*_w_ of about 2600. The FTIR spectrum of DPE-PBBB is shown in Figure 2. The major absorption bands attributed to the pentabromobenzene ring are 1317 cm^−1^ (semicircle stretching), 1061 cm^−1^ (C–Br stretching) and 552 cm^−1^ (ring bending). The bands at 1491 cm^−1^ and 1431 cm^−1^ (semicircle stretching) and 942 cm^−1^ (aromatic C–H bending) can be assigned to substituted diphenylethane. Solubility in organic solvents of DPE-PBBB was very poor, which did not allow for the performing of NMR analysis. The chemical structure of DPE-PBBB can be represented as follows. 
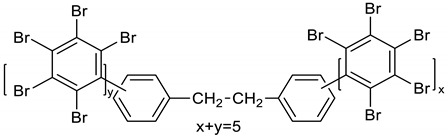
(2)

#### 2.11.3. Alkylation of Toluene with PBBB and TBXDB (T-PBBB-TBXDB)

Dichloroethane (300 mL), PBBB (84.8 g, 0.15 mol), TBXDB (66.1 g, 0.114 mol), and toluene (13.8 g, 0.15 mol) were placed in a 500 mL reactor fitted with a mechanical stirrer, thermometer, condenser, HBr trap, and N_2_ inlet. The mixture was heated to 60 °C. AlCl_3_ (1.3 g, 0.01 mol) was added by portions and the vigorous formation of HBr was observed. The mixture was kept at 60 °C for 2–4 h until the PBBB and TBXDB disappeared (controlled by GC and HBr evolution). The reaction mixture was washed three times with water (3 × 300 mL) and an aqueous solution of NaHCO_3_, with 30 min taken for each washing. The reaction mixture was cooled to 50 °C and the solid was filtered off and dried in an oven at 150 °C under reduced pressure for 24 h, giving 131.3 g, which corresponded to a ~98% yield. Based on the chemical analysis, the content of bromine was about 70%. Based on GPC the calculated molecular weight was *M*_n_ = 25,000 and *M*_w_ = 385,000 and polydispersity (PD) was PD = 15.4, *d*_n_/*d*_c_ = 0.09. An FTIR spectrum of T-PBBB-TBXDB is shown in Figure 3. The major absorption bands attributed to the pentabromobenzene ring are 1318 cm^−1^ (semicircle stretching), 1061 cm^−1^ (C–Br stretching) and 552 cm^−1^ (ring bending). A characteristic band at 1094 cm^−1^ (C–Br stretching) belongs to tetrabromoxylylene dibromide. The bands at 1432 cm^−1^ (semicircle stretching) and 936 cm^−1^ (aromatic C–H bending) are assigned to substituted toluene. Its chemical structure can be represented as follows. 
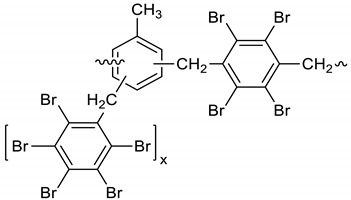
(3)

#### 2.11.4. Alkylation of Diphenylethane with PBBB and TBXDB (DPE-PBBB-TBXDB)

Dibromomethane (250 mL), PBBB (113.1 g, 0.2 mol), TBXDB (20.3 g, 0.035 mol), and diphenylethane (9.1 g, 0.05 mol) were placed in a 500 mL reactor fitted with a mechanical stirrer, thermometer, condenser, HBr trap, and N_2_ inlet. The mixture was heated to 60 °C. AlCl_3_ (0.8 g, 0.006 mol) was added by portions and the vigorous formation of HBr was observed. The mixture was kept at 60 °C for 2–4 h until the PBBB and TBXDB disappeared (controlled by GC and HBr evolution). The reaction mixture was washed three times with water (3 × 300 mL) and an aqueous solution of NaHCO_3_. After washing, the suspension of the product was added dropwise to isopropyl alcohol (1000 mL), stirred at 20 °C for 30 min, and the product filtered off. The weight of the product was 114.8 g, which corresponded to a ~94% yield. The content of bromine was about 75%. Based on GPC the calculated molecular weight was *M*_n_ = 19,000 and *M*_w_ = 134,000 and polydispersity was PD = 6.9, *d*_n_/*d*_c_ = 0.08. An FTIR spectrum of DPE-PBBB-TBXDB is shown in Figure 4. The major absorption bands attributed to the pentabromobenzene ring are 1318 cm^−1^ (semicircle stretching), 1061 cm^−1^ (C–Br stretching) and 552 cm^−1^ (ring bending). The characteristic band at 1094 cm^−1^ (C–Br stretching) belongs to tetrabromoxylylene dibromide. The bands at 2981 cm^−1^ (CH_2_ stretching), 1500 cm^−1^ and 1432 cm^−1^ (semicircle stretching) and 940 cm^−1^ (aromatic C–H bending) are assigned to substituted diphenylethane. The following structure was assigned to this product.

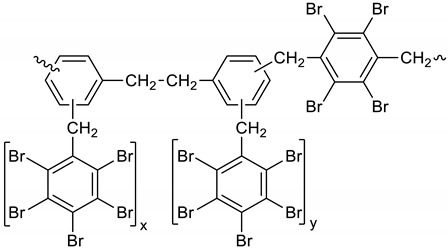
(4)

#### 2.11.5. Alkylation of PPE with PBBB (PPE-PBBB)

Polyphenylene ether oligomer (30 g), PBBB (366.4 g), and dibromomethane (700 mL) were placed in a 1 L reactor fitted with a mechanical stirrer, thermometer, condenser, HBr trap, and N_2_ inlet. The mixture was heated to 70 °C and AlCl_3_ (3.5 g) was added by portions. The mixture was stirred at 90 °C until the PBBB disappeared (5–6 h). The reaction mixture was washed at 50 °C with water (250 mL) and sodium bisulfite (2.5 mL, ~28%), water (250 mL), 5% Na_2_CO_3_ (250 mL), and water (250 mL), with 20 min taken for each washing. The solvent (~150 mL) was evaporated under reduced pressure. The remainder, heated at 40–50 °C, was added dropwise to isopropyl alcohol (450 mL) at 50 °C, over 1 h, under vigorous stirring. The mixture was stirred at 50 °C for 1 h and then cooled to 18 °C. The solid was filtered off and washed with isopropyl alcohol (150 mL) on the filter. The solid was dried in the oven at 105 °C, and at 150 °C under reduced pressure, for 3 and 7 h respectively, to give a white powder product (318 g, corresponding to ~92% yield, based on PBBB). The content of bromine was about 75%, which corresponds to about 2.1 PBBB group per one phenylene ether unit. The estimated molecular weight is *M*_w_ = 6,200. An FTIR spectrum of PPE-PBBB is shown in Figure 5. The major absorption bands attributed to the pentabromobenzene ring are 1317 cm^−1^ (semicircle stretching), 1061 cm^−1^ (C–Br stretching), and 553 cm^−1^ (ring bending). The bands at 1432 cm^−1^ (semicircle stretching) and 944 cm^−1^ (aromatic C–H bending) can be assigned to substituted polyphenylene ether. The chemical structure of PPE-PBBB can be represented as follows.

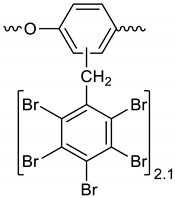
(5)

#### 2.11.6. Alkylation of PS with PBBB (PS-PBBB)

Dibromomethane (1.8 L) and polystyrene (104.15 g) were placed in a 2 L reactor fitted with a mechanical stirrer, thermometer, condenser, HBr trap, and N_2_ inlet. The mixture was heated to 66 °C to give a clear solution. PBBB (565.5 g) was then dissolved in the solution. AlCl_3_ (3.1 g) and SnCl_4_ (4.8 g) were added and the mixture was heated to 84 °C, after which the vigorous evolution of HBr started. The mixture was heated at 80 °C for about 6 h until the PBBB disappeared (by GC and HPLC). The reaction mixture was washed three times; the mixture was washed with water (1.5 L), saturated NaHCO_3_ solution (1.5 L) to give pH = 7, and again with water (1.5 L), with 30 min taken for each washing. After this, the reaction mixture was added dropwise to acetone (6 L) to induce precipitation. The reaction mixture was cooled to 20 °C and the solid was filtered off and dried in an oven at 105 °C under reduced pressure for 12 h to give 509 g, which corresponded to an ~86% yield, based on PBBB. According to elemental analysis, the content of bromine was about 68%, corresponding to about one PBBB molecule per aromatic ring, as shown in the structure below. Based on GPC, the calculated molecular weight was *M*_n_ = 19,300 and *M*_w_ = 40,100 and polydispersity was PD = 2.1. An FTIR spectrum of PS-PBBB is shown in Figure 6. The major absorption bands attributed to the pentabromobenzene ring are 1318 cm^−1^ (semicircle stretching), 1060 cm^−1^ (C–Br stretching) and 552 cm^−1^ (ring bending). The bands at 2981 cm^−1^ (CH_2_ stretching), 1499 cm^−1^ and 1432 cm^−1^ (semicircle stretching), and 941 cm^−1^ (aromatic C–H bending) can be assigned to substituted polystyrene. The chemical structure of PS-PBBB can be represented as follows.

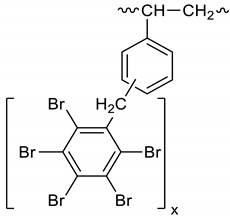
(6)

## 3. Results and Discussion

### 3.1. Thermogravimetry

TGA curves of two flame retardants made by alkylation with TBBB of toluene and diphenylethane are shown in Figure 7a,b. It is noticeable that degradation starts at about 370–375 °C and initially proceeds very quickly with a sharp maximum in the DTG curve. The second step of decomposition is much slower, and it extends all the way up to 600–650 °C. Eventually, the rate of the decomposition slows down at 650–700 °C and both flame retardants leave about 20% of solid residue behind. Since both T-TBBB and DPE-TBBB contain about 75% Br it is logical to assume that the first and the second steps of weight loss are associated with elimination of bromine along with some aromatic fragments. The TGA of two polymeric flame retardants made by alkylation of toluene and diphenylethane with TBBB and TBXDP are shown in Figure 7c,d. Performance of T-PBBB-TBXDP is very similar to that of T-TBBB (Figure 7a) but T-PBBB-TBXDP leaves higher solid residues of 24%. On the other hand, DPE-PBBB-TBXDP shows a more complex decomposition pattern with the shoulders on the DTG curve before and after the maximum weight loss. Nevertheless, the main pattern remains the same with a relatively fast debromination followed by a slower and more prolonged debromination. Finally, the TGA of the oligomer (PPE) and true polymer (PS) alkylated with TBBA are shown in Figure 7e,f. PPE-TBBB shows very fast initial decomposition with relatively low weight loss of only about 20%, which is followed by a much slower and prolonged debromination. By contrast, PS-TBBB has a slow initial decomposition step of about 50%, which is followed by an even slower step.

Table 2 summarizes TGA data in terms of the temperature of initial weight loss (*T*_5%_), main step of weight loss (maximum rate of weight loss (*T*_max_), approximate end of the first fast step (*T*_end_), and approximate weight lost in this step (wt. loss)), and solid residue at 700 °C. It is important to note that all of the newly synthesized polymeric flame retardants had high thermal stability, meaning they are stable enough to be processed even with high melting engineering resins like PA 6.6. Interestingly, the non-polymeric flame retardants and polymeric ones prepared by two different methods showed a similar decomposition pattern which is indicative of similar processes taking place, so we can reasonably speculate that the initial fast decomposition can be probably attributed to the elimination of pendant TBBB groups (dealkylation). It is also reasonable to assume that such fragments volatilize in the flaming conditions and quickly deliver a significant amount of bromine to the gas phase.

### 3.2. Combustion

#### 3.2.1. Polypropylene Copolymer

Polypropylene and its copolymers find significantly more applications in molded parts compared to other polyolefins [21]. Some of these molded parts need to be flame retarded; examples include electrical parts (wire nuts, lamp sockets, coil bobbins, connectors, wire, and cables), housing of electrical appliances, TV yokes, pipes for water discharge, fibers for textile applications, film, and sheets for roofing [22]. The relatively high fuel content of polypropylene (PP) combined with poor compatibility of common brominated flame retardants makes formulation of flame-retardant PP a significant challenge [23]. In order to reach a UL-94 V-0 rating in polypropylene, about 30–40% of flame retardant is required compared to 12–25% flame retardant loading for styrenic polymers or engineering thermoplastics [24]. In many molding applications copolymers of propylene and ethylene are used. These copolymers provide better impact strength but they are even more flammable than standard polypropylene.

Newly synthesized brominated flame retardants were tested in unfilled and talc-filled polypropylene-ethylene copolymer. Their compositions, flammability performance, and physical properties are shown in Table 3. Reference data for non-polymeric commercial flame retardant decabromodiphenylethane (FR-1410) and polymeric commercial flame retardant poly (pentabromobenzylacrylate) (FR-1025) are also shown in the table. FR-1410 has a high bromine content of 82% and because of this it requires lower loading compared to the new experimental flame retardants. Overall, FR-1410 provides good performance in PP copolymer, but it is not polymeric or a high molecular weight flame retardant. FR-1025 is truly polymeric, but it has a high glass transition temperature and is therefore too stiff for PP. As a result, it has a high heat distortion temperature but poor elongation at break.

Non-polymeric flame retardant DPE-PBBB required a slightly higher loading in terms of bromine content (22 versus 21) compared to FR-1410. DPE-PBBB showed a higher heat distortion temperature compared to FR-1410 whereas other mechanical properties were in line with the commercial benchmark. Polymeric PPE-PBBB required lower bromine loading (19.3 versus 23) compared to FR-1025. All newly synthesized polymeric flame retardants showed significantly better elongation at break compared to FR-1025. In the talc-filled compositions both DPE-PBBB and PS-PBBB showed comparable physical properties to those of FR-1025, but again elongation at break was significantly better.

#### 3.2.2. ABS and HIPS

Polystyrene and its copolymers have the tendency to depolymerize when exposed to fire temperatures and the volatile products are materials of high fuel value, namely, styrene monomer, styrene dimers, and related hydrocarbons such as benzene, lower-alkylbenzenes, and a few percent of oxygen-containing related aromatics [25]. Polystyrenes, unless blended with char formers, form little or no char by themselves [26]. The volatiles burn with copious soot formation. Moreover, while depolymerization is taking place, melt flow and drips also occurr, and the drips may be capable of igniting other flammable objects. For many uses of styrenics, especially electrical equipment, the requirements are to prevent a small source of ignition, such as a hot or sparking wire, from igniting the item, or, if the polymer is ignited, to cause it to self-extinguish quickly. Hence, the UL-94 test is the most common and to achieve a V-0 rating it is recommended to add a small amount of polytetrafluoroethylene (PTFE).

Some newly synthesized flame retardants were tested in ABS and compared with commercial flame retardants FR-1410 and F-3020. The compositions, flammability performance, and physical properties are shown in Table 4. The two commercial products had overall good physical properties but also had some deficiencies. FR-1410 has a low resin flow which limits its application in large and thin wall molded parts, whereas F-3020 shows excellent resin flow but poor elongation at break which makes its molded parts somewhat brittle. With a bromine/Sb_2_O_3_ ratio of 2.5 the newly synthesized flame retardants show similar flame retardant efficiency as the commercial flame retardants, e.g., they provide a V-0 rating at 10.0% of bromine in the formulation. Although most of the physical properties were found to be in line with those of commercial flame retardants, the impact strength was somewhat lower. In order to increase the impact strength, the loading of DPE-PBBB was increased to a level of 13% bromine and PPE-PBBB to a level of 11.5% and 13% bromine, but antimony trioxide was significantly decreased to a level of 2.1% and 1.5%. Although the newly synthesized flame retardants did not display an outstanding performance, they had an overall balance of high flame retardant efficiency and good physical properties.

The polymeric product of alkylation of diphenylethane with PBBB and TBXDB (DPE-PBBB-TBXDB) was tested for flammability and physical properties in HIPS. Results are presented in Table 5 with a comparison to commercial F-3014. As it is seen, 10% bromine and 4.0% antimony trioxide gave only a V-1 rating for both additives. However, interestingly an increase of bromine content to 11% and a decrease of antimony trioxide content to 2.1% resulted in a V-0 rating. A further decrease of antimony trioxide to 1.5% required a boost in the bromine content to 13%. Reference brominated epoxy polymer F-3014 showed a very similar flame retardant efficiency as DPE-PBBB-TBXDB. In terms of physical properties F-3014 had a higher impact and tensile strength, elongation at break and resin flow (MFI). On the other hand, experimental DPE-PBBB-TBXDP showed a somewhat higher tensile modulus and significantly higher heat distortion temperature (HDT) which is beneficial for some HIPS applications.

#### 3.2.3. Polyamide 6.6

Fire retardancy of aliphatic polyamides is mostly required in electric industries with typical applications in electrical connectors, terminal blocks, small electrical housings, clip fasteners, switch components, wire ties, and many other industrial parts [27]. Among other engineering resins, polyamides are known as materials with relatively high strength, high ductility, excellent resistance to short-term heat exposure, and good resistance to chemical solvents [28]. It is not easy to flame retard polyamides because many FRs cause destabilization (discoloration and reduction in melt viscosity) because of the polymer sensitivity to even low concentration of acids. Another problem encountered with some flame-retarded polyamides is the migration of the additive during processing, water conditioning, or long use of the polymer, which might lead to a loss of flame retardancy.

Newly synthesized brominated flame retardants were tested in glass-filled polyamide 6.6 and compared with commercial brominated polystyrene, FR-803P (Table 6). It is interesting to note that DPE-PBBB required a significantly lower loading of 10% Br and 4.8% Sb_2_O_3_ to achieve a V-0 rating compared to other experimental flame retardants as well as commercial FR-803P. Only the Izod impact strength of commercial FR-803P was somehow higher than the newly synthesized flame retardants. In general, the experimental flame retardants showed a noticeable advantage in the melt flow, which could be related to a lower loading because in general the experimental flame retardants were observed to have a higher bromine content (68–75%) compared to FR-803P (66%). All other properties of commercial FR-803P and the newly synthesized flame retardants were similar.

## 4. Conclusions

In this work, two new non-polymeric brominated flame retardants were successfully synthesized in one simple step by the alkylation of toluene or diphenylethylene with pentabromobenzyl bromide. Two polymeric brominated flame retardants were synthesized when monofunctional pentabromobenzyl bromide and difunctional tetrabromoxylylene dibromide were applied together to alkylate toluene or diphenylethylene. Finally, two other polymeric flame retardants were synthesized by alkylation of polyphenylene ether or polystyrene with pentabromobenzyl bromide. Chemical analysis showed that the new brominated flame retardants had bromine content in the range 68–75%. All new flame retardants showed good thermal stability in excess of 360 °C and therefore can be used in commodity resins as well as engineering thermoplastics. The new brominated flame retardants were tested in unfilled and mineral-filled polypropylene copolymer, HIPS, ABS, and glass-filled polyamide 6.6. In terms of efficiency, due to the high bromine content the new flame retardants performed similarly to or better than commercial flame retardants. For example, non-polymeric flame retardant DPE-PBBB performed well in glass-filled polyamide 6.6, where it showed a higher flame retardant efficiency compared to commercial brominated polystyrene. Two polymeric flame retardants DPE-PBBB-TBXDB and PPE-PBBB enabled low antimony V-0 formulations in HIPS and ABS, respectfully. Pentabromobenzyl bromide alkylated polystyrene surprisingly showed better flame retardant efficiency in polypropylene compared to all other newly synthesized flame retardants. The physical properties of the flame-retardant thermoplastics with all new flame retardants were in line with or better than the commercial references.

## 5. Patents

Two patents US 10.336,858 (2 July, 2019) and US 10,472,462 (12 November, 2019), both to Bromine Compounds, resulted from the work reported in this paper.

## Figures and Tables

**Figure 1 polymers-12-00352-f001:**
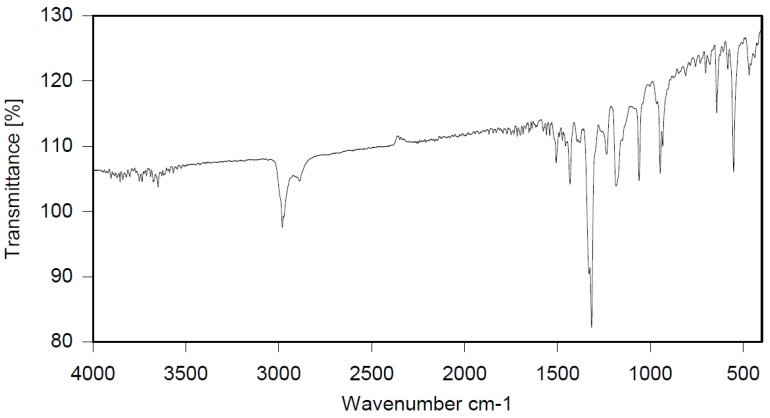
FTIR spectrum of toluene alkylated with entabromobenzyl bromide (T-PBBB).

**Figure 2 polymers-12-00352-f002:**
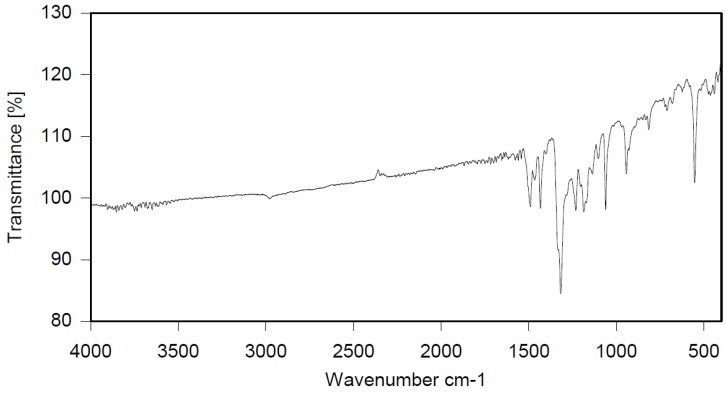
FTIR spectrum of diphenylethane alkylated with PBBB (DPE-PBBB).

**Figure 3 polymers-12-00352-f003:**
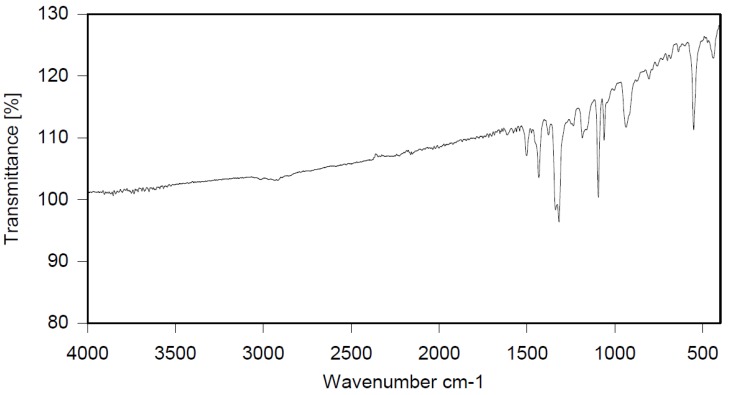
FTIR spectrum of toluene alkylated with PBBB and tetrabromoxylylene dibromide (T-PBBB-TBXDB).

**Figure 4 polymers-12-00352-f004:**
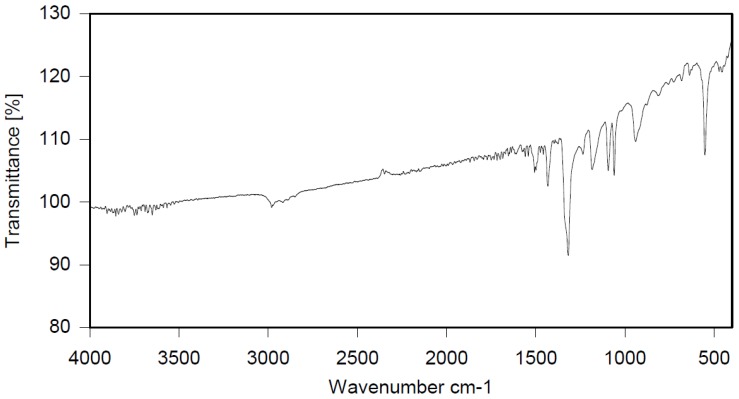
FTIR spectrum of diphenylethane alkylated with PBBB and TBXDB (DPE-PBBB-TBXDB).

**Figure 5 polymers-12-00352-f005:**
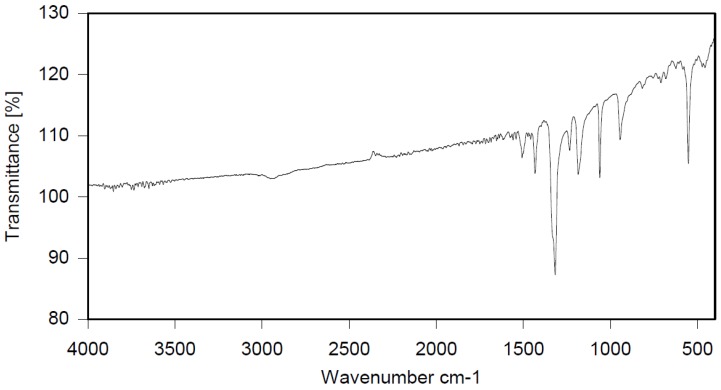
FTIR spectrum of polyphenylene ether alkylated with PBBB (PPE-PBBB).

**Figure 6 polymers-12-00352-f006:**
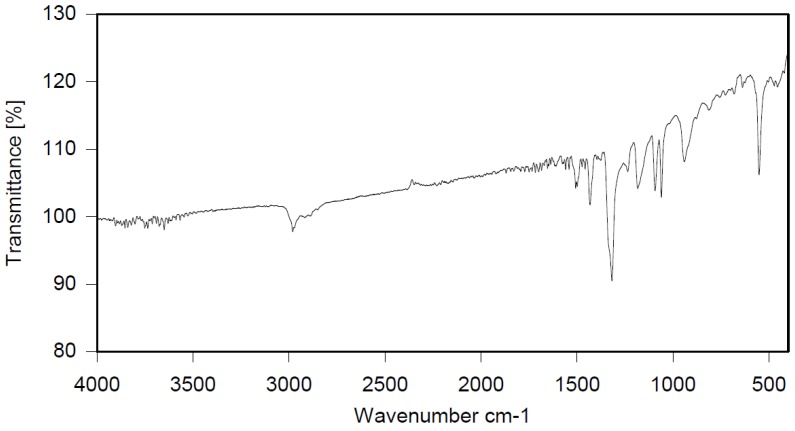
FTIR spectrum of polystyrene alkylated with PBBB (PS-PBBB).

**Figure 7 polymers-12-00352-f007:**
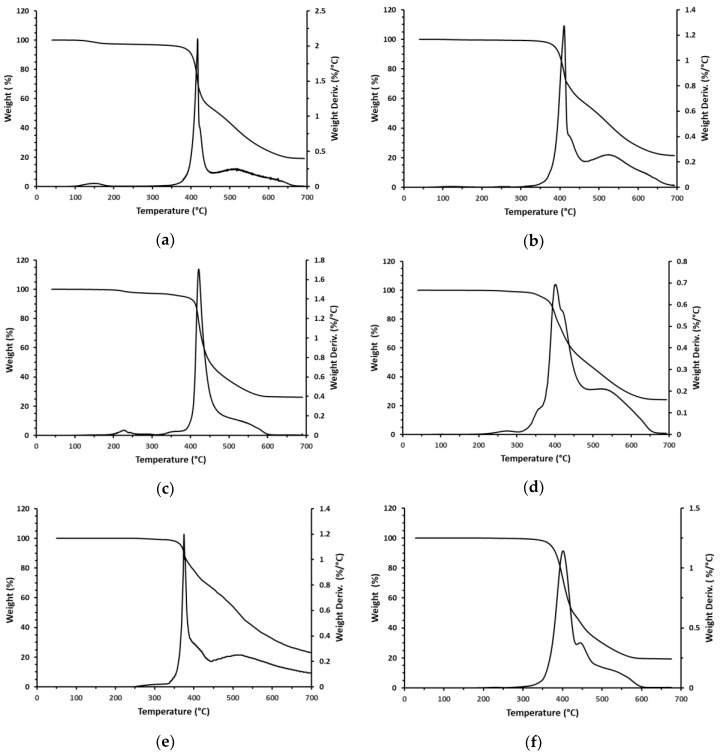
Thermogravimetric weight loss (TGA) and the rate of weight loss (DTG) of brominated flame retardants. (**a**) T-PBBB; (**b**) DBE-PBBB; (**c**) T-PBBB-TBXDP; (**d**) DPE-PBBB-TBXDP; (**e**) PPE-PBBB; (**f**) PS-PBBB.

**Table 1 polymers-12-00352-t001:** Polymers, reference flame retardants, synergists, fillers, stabilizers, and processing aids used in preparation of the flame-retardant plastics.

Component	Trade Name and Manufacture	Function
Impact-modified polypropylene (PP)	Capilene^®^ SL-50 (Caol)	Polymer matrix
Polyamide 6.6 (contains nucleating agent, mold release agent, and lubricant) (PA 6.6)	Aculon^®^ S 223D (DSM)	Polymer matrix
High-impact polystyrene (HIPS)	Styron^TM^ 1200 (Dow)	Polymer matrix
Acrylonitrile–butadiene–styrene copolymer (ABS)	ABS Magnum^TM^ 3404 (Styron)	Polymer matrix
Decabromodiphenylethane	FR-1410 (ICL-IP)	Reference flame retardant
Poly(pentabromobenzyl acrylate)	FR-1025 (ICL-IP)	Reference flame retardant
Brominated epoxy polymer	F-3014 (ICL-IP)	Reference flame retardant
Tribromophenol-end-capped brominated epoxy polymer	F-3020 (ICL-IP)	Reference flame retardant
Brominated polystyrene	FR-803P (ICL-IP)	Reference flame retardant
Antimony trioxide masterbatch (MB) containing 80 wt.% Sb_2_O_3_	FR00112 (Kafrit)	Flame retardant (FR) synergist
Polytetrafluoroethylene (PTFE)	Hostaflon^®^ 2017 (Dyneon)	Anti-dripping agent
Styrene–butadiene–styrene block copolymer (SBS)	SBS 501S (LG Chem)	Impact modifier
Talc	Lotalc	Filler
Masterbatch of talc, 60 wt.%	Talc MB (Kafrit)	Filler
Glass fibers	GF ChopVantage^®^ 3660 (PPG)	Filler
Blend of tris(2,4-ditert-butylphenyl)phosphite and pentaerythritol tetrakis [3-[3,5-di-tert-butyl-4-hydroxyphenyl]propionate] (50:50)	Irganox^®^ B 225 (BASF)	Antioxidant and heat stabilizer
Multifunctional nitrogen-containing hindered phenol	Acrawax^®^ C (Lonza)	Antioxidant and heat stabilizer
*N*,*N*′ ethylene bis stearamide	Irganox^®^ B1171 (BASF)	Lubricant
Calcium stearate	Ca-stearate	Lubricant

**Table 2 polymers-12-00352-t002:** TGA data of brominated flame retardants.

Flame Retardant	Initial Decomposition	Main Step of Weight Loss			Solid Residue
	*T*_5%_, °C	*T*_max_, °C	*T*_end_, °C	Wt. loss, %	Wt.%
T-PBBB	370	410	450	35	19
DBE-PBBB	375	405	455	35	21
T-PBBB-TBXDP	375	410	475	55	26
DBE-PBBB-TBXDP	360	400	480	50	24
PPE-TBBB	370	375	390	20	23
PS-TBBB	360	410	435	50	19

**Table 3 polymers-12-00352-t003:** Flammability performance and physical properties of impact-modified PP. Legend: HDT, heat distortion temperature; MFI, melt flow index.

* Composition, wt.%	1	2	3	4	5	6	7	8	9	10
Flame Retardant	FR-1410	FR-1025	DPE-PBBB	DPE-PBBB-TBXDB	PPE-PBBB	PS-PBBB	FR-1025	DPE-PBBB	FR-1025	PS-PBBB
Impact-modified PP	68.5	54.6	56.7	54.7	54.6	59.7	51.9	53.3	50.3	49.5
Flame retardant	25.6	32.4	29.3	30.7	30.9	28	25.4	24.0	25.3	26.1
Talc or talc MB							15.0	15.0	16.7 (10)	16.7 (10)
Antimony trioxide MB	13.1	12.8	13.8	14.4	14.4	12.1	7.5	7.5	7.5	7.5
Br content, % calculated	21.0	23.0	22.0	23.0	23.0	19.3	18.0	18.0	18.0	18.0
Sb_2_O_3_, % calculated	10.5	11.5	11.0	11.5	11.5	9.7	6.0	6.0	6.0	6.0
Br/Sb_2_O_3_ ratio	2.0	2.0	2.0	2.0	2.0	2.0	3.0	3.0	3.0	3.0
UL-94 rating	V-0	V-0	V-0	V-0	V-0	V-0	V-0	V-0	V-0	V-0
Impact strength (J/m)	72	31	27	35	32	33	26	28	35	34
Tensile strength (MPa)	21.2	24.6	17.2	17.6	17.0	18.9	22.3	18.0	20.4	18.6
Elongation at break (%)	127	6.1	64	27	102	47	6.9	27	27	57
Tensile modulus (MPa)	1550	1800	1490	1660	1450	1620	2080	2060	2170	1790
HDT (°C)	55	84	63	58	57	61	77	67	74	61
MFI (g/10 min)	5.3	12.5	4.1	3.8	5.9	3.8	6.2	3.5	5.0	2.9

* All formulations contained 0.2 wt.% Irganox B 225.

**Table 4 polymers-12-00352-t004:** Flammability performance and physical properties of ABS plastic.

* Composition, wt.%	1	2	3	4	5	6	7	8
Flame Retardant	FR-1410	F-3020	DPE-PBBB	DPE-PBBB	DPE-PBBB-TBXDB	PPE-PBBB	PPE-PBBB	PPE-PBBB
ABS	82.5	76.8	81.4	80.4	81.4	81.4	81.7	80.5
Flame retardant	12.2	17.9	13.3	17.4	13.3	13.3	15.3	17.3
Antimony trioxide MB	5.0	5.0	5.0	1.9	5.0	5.0	2.6	1.9
Br content, % calculated	10.0	10.0	10.0	13.0	10.0	10	11.5	13.0
Sb_2_O_3_, % calculated	4.0	4.0	4.0	1.5	4.0	4.0	2.1	1.5
Bromine/Sb_2_O_3_ ratio	2.5	2.5	2.5	8.7	2.5	2.5	5.5	8.7
UL-94 rating	V-0	V-0	V-0	V-0	V-0	V-0	V-0	V-0
Impact strength (J/m)	128	113	71	104	69	92	78	74
Tensile strength (MPa)	40	43	37	39	38	38	38	37
Elongation at break (%)	16	3	7	13	11	6	6	6
Tensile modulus (MPa)	2400	2330	2330	2160	2330	2080	2100	2070
HDT (°C)	78	76	77	78	78	77	78	79
MFI (g/10 min)	8.8	22.7	15.3	11.7	13.4	12.4	12.3	10.6

* All formulations contained 0.1 wt.% PTFE and 0.2 wt.% Irganox B 225.

**Table 5 polymers-12-00352-t005:** Flammability performance and physical properties of HIPS.

* Composition, wt.%	1	2	3	4	5	6
Flame Retardant	F-3014	F-3014	F-3014	DPE-PBBB-TBXDB	DPE-PBBB-TBXDB	DPE-PBBB-TBXDB
HIPS	78.0	78.8	76.1	82.2	83.3	81.5
Flame retardant	16.7	18.3	21.7	12.5	13.8	16.3
Antimony trioxide MB	5.0	2.6	1.9	5.0	2.6	1.9
Br content, % calculated	10	11	13	10	11	13
Sb_2_O_3_, % calculated	4.0	2.1	1.5	4.0	2.1	1.5
Bromine/Sb_2_O_3_ ratio	2.5	5.2	8.7	2.5	5.2	8.7
UL-94 rating	V-1	V-0	V-0	V-1	V-0	V-0
Impact strength (J/m)	67	70	67	43	43	38
Tensile strength (MPa)	26	25	25	24	25	23
Elongation at break (%)	40	57	47	17	15	11
Tensile modulus (MPa)	1980	1920	1910	2100	2070	2050
HDT (°C)	65	66	65	71	72	72
MFI (g/10 min)	14.4	13.8	14.2	4.2	4.0	4.2

* All formulations contained 0.1 wt.% PTFE and 0.2 wt.% Irganox B 225.

**Table 6 polymers-12-00352-t006:** Flammability performance and physical properties of glass-filled polyamide 6.6.

* Composition, wt.%	1	2	3	4	5
Flame Retardant	FR-803P	T-PBBB	DPE-PBBB	PPE-PBBB	PS-PBBB
PA 6.6	43.5	46.3	51.3	45.8	44.1
Glass fiber	30	30	30	30	30
Flame retardant	19.7	16.9	13.3	17.4	19.1
Antimony trioxide MB	6.2	6.2	4.8	6.2	6.2
Br (calculated)	13	13	10	13	13
Sb_2_O_3_ (calculated)	4.8	5.0	3.8	5.0	5.0
Br/Sb_2_O_3_ (calculated)	2.7	2.6	2.6	2.6	2.6
UL-94 rating	V-0	V-0	V-0	V-0	V-0
Izod Impact (J/m)	111	104	103	104	97
Tensile strength (MPa)	143	150	150	135	137
Elongation at break (%)	3.3	4.2	4.7	4.3	3.5
Tensile modulus (MPa)	10050	10750	9950	8630	10450
HDT (°C)	225	226	231	228	216
MFI (g/10 min)	7	11	29	13	15

* All formulations contained 0.2 wt.% Acrawax C, 0.2 wt.% Irganox 51171, and 0.2 wt.% Ca-stearate.

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
