# Peer review of "Alkylation of Aromatic Compounds with Pentabromobenzyl Bromide and Tetrabromoxylene Dibromide as a New Route to High Molecular Weight Brominated Flame Retardants"

_polymers, 2020, doi:10.3390/polym12020352_

Round 1

Reviewer 1 Report

In the article entitled: "Alkylation of Aromatic Compounds with Pentabromobenzyl Bromide and Tetrabromoxylene Dibromide as a New Route to High Molecular Weight Brominated Flame Retardants", prepared by authors: Mark Gelmont, Michael Yuzefovitch, David Yoffe, Eyal Eden and Sergei Levchik it was shown, that two new non-polymeric brominated flame retardants were successfully synthesized in one simple step by the alkylation of toluene or diphenylethylene with pentabromobenzyl bromide to function as new flame retardants in polymeric systems such as polypropylene copolymer, HIPS, ABS and glass-filled polyamide 6.6.

It was proved that in terms of efficiency, due to the high bromine content the new flame retardants performed similar to or better than commercial flame retardants.

The pape is interesting and the topic is important. The paper sounds good and in my opinion can be published without changes.

Author Response

Thank you for your review

Reviewer 2 Report

This work seems to be a technical report rather than a research paper. It lacks mechanism studies. On the other hand, this work is from the patents, US 10.336,858 (July 2, 2019) and US 10,472,462 (November 12, 2019). As a result, I don't recommend this paper to be published in Polymers.

Author Response

Thank you for your review.

In fact this work was performed in industrial laboratory. However, we believe it will be of interest for Polymers audience because it reports on many practical aspects.

Reviewer 3 Report

In this work, high molecular weight and polymeric brominated flame retardants was synthesized through electrophilic C-alkylation of different aromatic compounds including pentabromobenzyl bromide (PBBB) or tetrabromoxylylene dibromide (TBXDB), which were then added to modify different thermoplastic. This work was of some significance in consideration of the environment. Besides, the performance in terms of flame retardancy and mechanical properties of thermoplastic modified by DPE-PBBB, DPE-PBBB-TBXDB, PPE-PBBB, PS-PBBB, etc., were comparable or better than the reference commercial flame retardants. Therefore, considering that the synthesized flame retardants were useful for the practical application and the reported results were good, this work could be accepted under some revision.

Line 158, Page 5, the authors referred to “Solubility in organic solvents of P-PBBB was very poor which didn’t allow to perform NMR analysis.” Why not choosing solid-state NMR analysis? And there are no FTIR analyses of flame retardants. The authors did not introduce all the newly synthesized to modify thermoplastic, why? For example, in ABS plastic, only DPE-PBBB, DPE-PBBB-TBXDB and PPE-PBBB were used, while not for PS-PBBB, T-PBBB, and T-PBBB-TBXDB.

Author Response

Thank you for your review and comments. 

FTIR spectra and their discussion was added to the paper. Sorry, we don't have solid state NMR available in our institution. The work reported in this paper was a part of much larger project. Selection of the flame retardants for specific polymers was based on much larger body of experiments which we cannot report here.